# Malignant Peripheral Nerve Sheath Tumors: Latest Concepts in Disease Pathogenesis and Clinical Management

**DOI:** 10.3390/cancers15041077

**Published:** 2023-02-08

**Authors:** Chengjun Yao, Haiying Zhou, Yanzhao Dong, Ahmad Alhaskawi, Sohaib Hasan Abdullah Ezzi, Zewei Wang, Jingtian Lai, Vishnu Goutham Kota, Mohamed Hasan Abdulla Hasan Abdulla, Hui Lu

**Affiliations:** 1Department of Orthopedics, The First Affiliated Hospital, Zhejiang University, #79 Qingchun Road, Hangzhou 310003, China; 2School of Medicine, Zhejiang University, #866 Yuhangtang Road, Hangzhou 310058, China; 3Department of Nuclear Medicine, The First Affiliated Hospital, Zhejiang University, #79 Qingchun Road, Hangzhou 310003, China; 4Department of Orthopaedics, Third Xiangya Hospital, Central South University, #138 Tongzipo Road, Changsha 410013, China; 5Alibaba-Zhejiang University Joint Research Center of Future Digital Healthcare, Zhejiang University, #866 Yuhangtang Road, Hangzhou 310058, China

**Keywords:** malignant peripheral nerve sheath tumor, neurofibromatosis type 1, plexiform neurofibroma, atypical neurofibromatous neoplasm of unknown biological potential, molecular diagnosis, target therapy

## Abstract

**Simple Summary:**

Malignant peripheral nerve sheath tumor (MPNST) is a soft tissue sarcoma with limited therapeutic interventions and a poor prognosis. This review summarized the current understanding of the pathogenic mechanisms behind MPNST and the latest concepts in clinical management from diagnosis to therapeutic intervention. Additionally, the developments in molecular diagnosis and targeted therapies for MPNST are highlighted. It concluded with the challenges and prospects of MPNST management.

**Abstract:**

Malignant peripheral nerve sheath tumor (MPNST) is an aggressive soft tissue sarcoma with limited therapeutic options and a poor prognosis. Although neurofibromatosis type 1 (NF1) and radiation exposure have been identified as risk factors for MPNST, the genetic and molecular mechanisms underlying MPNST pathogenesis have only lately been roughly elucidated. Plexiform neurofibroma (PN) and atypical neurofibromatous neoplasm of unknown biological potential (ANNUBP) are novel concepts of MPNST precancerous lesions, which revealed sequential mutations in MPNST development. This review summarized the current understanding of MPNST and the latest consensus from its diagnosis to treatment, with highlights on molecular biomarkers and targeted therapies. Additionally, we discussed the current challenges and prospects for MPNST management.

## 1. Definition and Epidemiology

Malignant peripheral nerve sheath tumor (MPNST) is a relatively rare tumor, accounting for 5–10% of all soft-tissue sarcomas [1,2]. It refers to malignant tumors of the peripheral nerve or nerve sheath cells, but it excludes epineurium or nerve vasculature. Although MPNSTs usually originate from Schwann cells or pluripotent cells of the neural crest origin [3], pathological analysis reveals that other tissue types may potentially be involved in its composition, which is currently inconclusive [4]. The World Health Organization (WHO) categorized MPNST as a soft-tissue sarcoma for the first time in 2013. Subtypes of epithelioid MPNST, malignant triton tumor, and glandular MPNST were also described. The prevalence of MPNST in the general population is one in 100,000, affecting both genders equally [5]. Notably, 50% of the MPNST patients had neurofibromatosis type 1, and 10% had a history of radiation exposure [5], the remaining being mostly sporadic [6]. The average age of onset is around 30–50 years, although, in NF1 patients, it may occur 10 years earlier on average. It occurs mainly in the proximal limbs, followed by the trunk, head, and neck. The main clinical manifestations are pain and numbness; however, they are not specific symptoms, and MPNSTs are difficult to distinguish from other nerve lesions [5,7,8,9,10]. MPNST is still difficult to diagnose and treat, and the overall prognosis is poor. Although MPNST is a rare disease, the mortality rate is high. The median survival time is largely based on subtypes of MPNST and molecular variations. MPNST development may also be influenced by genetic differences between ethnic groups [11].

## 2. Etiology and Risk Factors

### 2.1. Neurofibromatosis Type 1

Neurofibromatosis type 1 (NF1) is a complex autosomal dominant disorder characterized by various germline mutations and clinical manifestations in multiple organs [12]. The global average incidence of NF1 is 1/3000, but this varies by region due to the founder effect and de novo mutation factors [13]. *NF1* is located at 17q11 and has 14 protein-coding genes. It primarily encodes neurofibromin, an analog of the negative regulator of the RAS proto-oncogene expressed in a variety of tissues. It inactivates RAS by accelerating the conversion of active guanosine triphosphate (GTP) bound RAS to the inactive guanosine diphosphate (GDP) bound RAS [14]. *NF1* mutation can lead to abnormal cell growth mediated by MEK, AKT, and other downstream pathways [15]. A germline microdeletion of *NF1* and flanking loci affect 5–10% of NF1 patients [16]. *SUZ12* gene loss in *NF1* microdeletion is also involved in tumor formation [17]. The main clinical manifestations of NF1 include pigmented lesions, cafe-au-lait macules (Figure 1), skin fold freckles, Lisch nodules (pigmented iris hamartoma), dermal neurofibromas, and peripheral nerve tumors [18]. MPNST is the leading cause of premature death in NF1 patients [19]. Clinical symptoms are mainly used to diagnose NF1. Genetic testing of blood genomic DNA and mRNA, as well as fluorescent in situ hybridization, are important for diagnosing NF1 in patients with atypical clinical symptoms [12]. The *NF1* genotype–phenotype correlation is being highly elucidated due to the development of new analysis techniques, such as multiplex ligation-dependent probe amplification (MLPA), comparative genomic hybridization (CGH) array, and next-generation sequencing (NGS). MLPA, in particular, is an efficient method for diagnosing and classifying *NF1* microdeletions [20].

### 2.2. Radiation 

The risk of post-radiation sarcoma in patients who undergo radiation therapy has been reported to be about 0.06% [21]. Approximately 10–13% of MPNST patients have a history of therapeutic irradiation [22]. According to a systematic review, the average age of radiation-induced MPNST is 31.7 ± 18.2 years, and the latency following radiotherapy is 13.5 ± 7.8 years [23]. The median survival duration is 11 months, with a five-year survival rate of 6.8% [23], which is lower than in sporadic MPNST [23,24]. There have been few studies on the mechanism of radiation-induced MPNST, where they found that loss of H3K27me3 expression is a highly sensitive marker for both sporadic and radiation-induced MPNST [25], implying that the post-radiation MPNST may share the same mechanism pathway as sporadic MPNST to some extent.

### 2.3. Plexiform Neurofibroma and Atypical Neurofibromatous Neoplasm of Unknown Biological Potential

Several studies over the last decade have indicated that NF1-associated MPNSTs typically begin as plexiform neurofibroma (PN) and atypical neurofibromatous neoplasm of unknown biological potential (ANNUBP). As PN and ANNUBP are considered precancerous lesions, they can help illustrate the pathogenesis of MPNST. PN is a benign precursor lesion in about 50% of NF1 patients [26]. The probability of malignant progression to MPNST is 10–15% [27]. PN is commonly detected in childhood and grows rapidly during that period. Growth in adulthood is usually indicative of malignancy potential [28]. In clinical manifestations, it is difficult to distinguish PN from MPNST since they both exhibit similar neurological symptoms (pain, neurologic impairment, and motor dysfunction) and disfigurement [29]. Some PN patients remain asymptomatic throughout their lives [30]. ANNUBP is a newly defined NF1-associated tumor that manifests as nuclear atypia, hypercellularity, and increased mitotic activity; thus, its malignant potential is uncertain [31]. ANNUBP can increase FDG uptake in PET/CT and is strongly correlated with MPNST in pathological manifestations. However, the risk of recurrence is low, with no risk of metastasis [31]. Other nerve sheath tumors (Schwannoma, Ganglioneuroma) can present with malignant tumors, but many studies indicate a rare occurrence rate [32,33]. 

In summary, previous studies have found numerous risk factors associated with the onset of MPNST (Figure 2). NF1 is the most important factor in 50% of MPNST patients. The history of therapeutic irradiation can also increase the risk of MPSNT. Possession of PN and ANNUBP has malignant potential to cause MPNST development. Additionally, aging is an important risk factor because MPNST development takes a long time. 

## 3. Mechanisms of MPNST Pathogenesis

### 3.1. Genetic Mechanism

Significant breakthroughs have been made in understanding the genetic mechanisms of MPNST in the previous decade. Researchers now have access to more genetic data related to MPNST because of the application of genetic sequencing in clinical diagnosis. In the case of NF1-associated MPNST, sequential, multiple-hit genetic changes may eventually lead to the transformation of Schwann cells into precancerous lesions and MPNST. Knocking out the *NF1* from the cell (*NF1^flox/flox^* embryos) and mouse (*Nf1^flox/flox^; DhhCre*) models has been found to cause PNs similar to those seen in humans [34]. Simultaneously, the *NF1* mutation duplicates are significantly associated with the development of lesions [34]. However, only the loss of *NF1* cannot result in the malignant transformation of PN to MPNST, suggesting that other genes are involved in developing MPNST [35]. Loss of CDKN2A is a common alteration of ANNUBP and MPNST, which is unrelated to NF1-associated or sporadic causes [36,37]. The transition to the premalignant state is driven by the extent of *CDKN2A/B* deficiency [38]. A haploid mutation can lead to atypical neurofibroma, and the homozygous deletion is defined as ANNUBP [38]. Conditional ablation of *NF1* and *CDKN2A* in the Schwann cell lineage results in ANNUBP and can progress to MPNST in the *Postn-Cre Nf1^flox/flox^ Arf^flox/flox^* mouse model [39]. Function loss of polycomb repressive complex 2 (PRC2) is also associated with MPNST. Lee et al. found that dysfunction of PRC2 occurs in 70% of NF1-associated MPNST, 92% of sporadic MPNST, and 90% of radiotherapy-associated MPNST [17]. EED and SUZ12 are the core components of PRC2, which work together to methylate Lys27 of histone H3 to produce H3K27Me3, which can inhibit nerve repair genes [40]. Almost 80% of MPNST exhibited EED or SUZ12 deletion, resulting in the loss of PRC2 function and leading to H3K27Me3 global hypomethylation [17]. Researchers transferred wild type SUZ12 or EED into the MPNST cell line ST88-14 (which has lost H3K27Me3) to investigate the association between PRC2 and MPNST. After SUZ12 transfer, H3K27me3 levels recovered, and cell growth was significantly reduced [17]. Ma et al. found that EED knockdown can cause epigenetic changes by reducing H3K27me3; however, this effect was inhibited by the induction of CDKN2A. It is speculated that the co-mutation of PRC2 encoding genes and CDKN2A resulted in the progression of benign neurofibroma to MPNST [41]. The present findings suggest that mutations in NF1, CDKN2A, and genes encoding PRC2 complex proteins may play a synergistic role in the development of MPNST, particularly NF1-associated MPNST. Furthermore, the mutation order may also coordinate with the biomarker changes (Figure 3).

Several other genetic targets have been investigated concerning MPNST. Hirbe et al. illustrated that co-mutations of *NF1* and *TP53* can lead to MPNST in mouse models (*Gfap-Cre Nf1^flox/flox^*) without developing the PN phase [42]. This may indicate that *TP53* is an independent prognostic factor for MPNST. Holtkamp et al. found increased *EGFR* expression, decreased *ERBB2* expression, and decreased expression of the tumor suppressor gene *PTEN* in MPNST cells [43]. Animal experiments have also shown that only concurrent EGFR overexpression or *PTEN* deletion in the presence of *NF1* mutations results in 100% MPNST manifestation [43].

However, only a few studies have revealed a possible pathogenesis for sporadic de novo MPNST. Some evidence suggests that sporadic MPNST has the same mutant genes, but in a different order, which may lead to atypical pathological changes and various prognoses. For example, the somatic *NF1* mutation of sporadic MPNST is similar to NF1-associated MPNST [44]. *CDKN2A* or *PRC2* mutations have also been seen in sporadic MPNST [25]. However, recent evidence indicates that some point mutations, such as *BRAF* V600E and *NRAS* Q61, are detected in sporadic MPNST. However, none of these mutations have been found in patients with NF1-associated MPNST or post-radiation MPNST [45]. Another study of 201 MPSNTs demonstrated that 11.9% of *NF1*-wild type MPSNTs harbored *BRAF* mutations compared to 2.9% of *NF1*-altered MPNSTs. *CDKN2A* is significantly altered in both *NF1*- and *BRAF*-altered MPNSTs [46]. The genetic differences between NF1-associated MPNST and sporadic MPNST may reflect different oncogenetic pathways [47].

### 3.2. Signaling Pathway and Microenvironment

The cellular signaling pathways of MPNST and the tumor microenvironment are important areas of future research. Particularly, alterations in the *NF1* gene can result in abnormal activation of the RAS pathway, which can promote cell proliferation via the downstream RAF-MEK-ERK (MAPK) and PI3K-AKT-mTOR pathways. MEK is significantly activated in MPNST [48], and clinical evidence demonstrates that MEK inhibitors are significantly effective in PN [49] and MPNST [50]. The mTOR signaling pathway regulates cell growth, survival, proliferation, cytoskeletal organization, and autophagy [51]. Biplab Dasgupta et al. first reported that mTOR signaling is significantly activated in NF1-knockout cells and animals and that the rapamycin (mTOR inhibitor) can inhibit NF1-related tumor growth [15]. Johansson et al. found that everolimus inhibits the growth of 19–60% of NF1-associated or sporadic MPNST cell lines [52]; however, it is ineffective in clinical trials [53]. Wnt/β-catenin signaling is strongly associated with various human cancers [54]. A study showed that the gene expression of 20 components or regulators of the Wnt pathway are altered in MPNST compared to benign neurofibromas [55]. However, the precise role of the Wnt pathway in MPNST is still undetermined. Other signaling pathways including neuregulin-1/ErbB [56] and EGFR–STAT3 [57], which have also been reported in MPNST studies. In the MPNST microenvironment, an interesting finding is that the NF1+/− microenvironment may contribute to neurofibroma development [58]. It is possibly due to the alteration in immune cells, such as CD8+ T cells and natural killer cells, compared to wild-type counterparts [59]. NF1-heterozygous mast cells were also found to be activated by c-kit signaling and to promote tumor cell growth by releasing TGF-β [60], which leads to related clinical studies of tyrosine kinase receptors targeting KIT receptors [61]. Others, such as macrophages, are also involved in tumor formation [62]. Together, the alterations in cellular signaling pathways and the NF1+/− microenvironment promote cell survival and proliferation of MPNST (Figure 4).

## 4. Diagnosis

There are currently no highly accurate diagnostic criteria for MPNST. Due to its strong similarity in symptoms and radiological imaging with PN and other soft tissue tumors, it is difficult to distinguish MPNST from other soft tissue tumors. Therefore, effective clinical diagnostic criteria are urgently needed. The latest 2022 National Comprehensive Cancer Network (NCCN) clinical guidelines summarized the primary modalities of MPNST diagnosis. In addition to conventional imaging and pathological means, gene mutation analysis and molecular detection during the pathogenesis of MPNST are the latest methods for diagnosing MPNST. Applying new molecular targets will help to diagnose and grade MPNST in a better way.

### 4.1. Imaging

Magnetic resonance imaging (MRI) is one of the most commonly utilized imaging techniques for soft tissue sarcomas. With the advancement of MRI technology, its sensitivity and specificity in tumor diagnosis are constantly improving [63]. MRI images of MPNST revealed a low signal in T1WI and a strong signal in T2WI. Some unusual symptoms, such as invasion of fat planes, heterogeneity, ill-defined margins, and edema surrounding the lesion, have been linked to MPNST [64]. Broski et al. found that perilesional edema, cystic degeneration or necrosis, and irregular margins can attain 100% specificity when these three signs are met simultaneously [65]. Yun et al. devised a scoring system combining the clinical presentations and imaging characteristics to achieve 100% sensitivity and specificity in MPNST diagnosis [66], although further validation is still needed. A meta-analysis summarized the role of different MRI sequences in the differential diagnosis of MPNST and benign peripheral nerve sheath tumor (BPNST) and found that minimum apparent diffusion coefficient (ADCmin) alone could achieve nearly 100% sensitivity. In comparison, diffusion weighted imaging (DWI) and ADC sequences could achieve 95% specificity [63]. The studies above have proven the efficacy of MRI in MPNST diagnosis; however, it is seldom used as a gold standard in clinical practice.

The NCCN guidelines recommend using PET/CT for MPNST diagnosis, as MPNST usually results in a significant increase in 18F-FDG uptake. PET/CT is a well studied technique that is thought to be more sensitive than conventional MRI; however, its diagnostic cutoff value is debatable [65]. It is mainly believed that there is no possibility of malignancy when the standard uptake value (SUV) is less than 2.5, and MPNST should be addressed when the maximum standard uptake value (SUVmax) is greater than 3.5 [67]. However, some other studies suggest that SUVmax values greater than 4.0 [68] or 6.1 [69] have a higher specificity in MPNST diagnosis. Considering the poor prognosis of MPNST, the SUVmax cutoff of 3.5 is still commonly used [65,70]. PET-guided biopsies provide a higher diagnostic value for tumors with intermediate SUVmax (2.5–3.5) and higher detection sensitivity [71]. PET/CT can also detect metastasis in real-time, which is of great significance for the classification and prognosis of MPNST. It is worth noting that PET/CT and MRI serve complementary roles. However, there have been few studies on the effects of combining PET/CT and MRI, and only a few institutions are equipped to perform PET/MR; thus, additional research is needed [65,72,73,74].

### 4.2. Biopsy

A biopsy is strongly recommended for diagnosing and grading soft tissue sarcomas. However, performing a biopsy for suspected MPNST remains debatable. Core needle biopsy is a method using a hollow-bored needle (typically 18 gauge) to obtain tissue from a suspected tumor. Image-guided core-needle biopsy (IGCNBx) is a standard procedure for most soft tissue sarcomas, with a 90% accuracy rate and minimal risk of tumor seeding [75]. However, there is still apprehension about performing core needle biopsies in MPNST due to the risk of nerve injury and the unknown accuracy of MPNST differentiation [75]. Pianta et al. [76] reported an excellent correlation between core biopsy and excised surgical specimen histology, with 100% accuracy for CT-guided core biopsy. In their study, however, 60% of patients reported pain related to their lesion, and 12% experienced pain aggravation. Complications are associated with tumor size, depth, and distance between needle tips and traversing nerves [76]. Graham et al. reported that the accuracy in differentiating MPNST and BPNST is 94%, with no long-term complications, indicating the accuracy and safety of core biopsy in MPNST [75]. Additionally, fine needle aspiration (FNA) is performed in MPNST biopsy [77,78]. H3K27me3 immunohistochemistry can assist in distinguishing MPNST from cytomorphologic mimics in FNA specimens [78].

Furthermore, it is debatable to pursue a biopsy versus upfront resection of a suspected MPNST. A biopsy, rather than an upfront resection, may only focus on a confined tumor area, but MPNST tends to be mixed with precursory lesions or benign changes. Therefore, a biopsy may be less sensitive than upfront resection [79].

### 4.3. Pathology

There is no defined subset of MPNST markers for pathological analysis. A specific pathological criterion for MPNST, particularly sporadic MPNST, is lacking due to the heterogeneity of gene mutation loci and immunohistochemical manifestations. Therefore, distinguishing it from other soft tissue sarcomas is challenging. Currently, MPNST diagnosis is primarily based on exclusion. MPNST is a tumor of nerve origin, usually attached to a nerve trunk, is white, solid, fleshy, and sometimes has myxoid changes. The main distinction is to rule out other nerve tumors. Microscopically, the MPNST can have a variety of morphologies, and as a result it is often classified into distinct subtypes [80]. MPNST has elongated, cusped, curved, or wavy nuclei with scant cytoplasm. The nucleus can be hyperchromatic or vesicular, with coarse chromatin in the latter, and the cells tend to be obtuse and pointed, typical features of neurilemmal differentiated cells (Figure 5). Furthermore, cell necrosis, mitosis, and hemorrhage are common pathological changes in MPNST [80]. In general, these findings are nonspecific and cannot be used to diagnose MPNST, but they are helpful in the differential diagnosis. For example, diffuse cellularity and conspicuous mitoses may ensure the differentiation of MPNST from atypical neurofibroma and nonneurogenic tumors [81].

### 4.4. Immunohistochemistry

MPNST lacks distinct molecular markers. As previously stated, the immunohistochemistry analysis is also based on the diagnosis of exclusion. Some important markers are listed in Table 1 and Figure 6. S-100 is a critical immunohistochemical marker for Schwann-associated tumors. However, differentiation of Schwann cells in MPNST is often inadequate and variable, and the S-100 positivity rate in MPNST is only 50–60% [82]; therefore, it is nonspecific for MPNST diagnosis. The negativity of the S-100 protein may indicate de-differentiation of Schwann cells; consequently, the ratio of S-100 negativity may predict malignancy. One study showed that a lack of tumor S-100 immunoreactivity was associated with a five-fold increased risk of distant metastasis and a 3.45-fold increased risk of mortality [83]. Apart from that, diffused expression of S-100 should prompt consideration of other diagnoses, such as melanoma and cellular Schwannoma [80]. SOX-10 has similar sensitivity as S-100 in the diagnosis of neural crest-derived tumors. However, SOX-10 has poor sensitivity in MPNST diagnosis because of its variable expression [84]. Some other markers are also conventionally used in MPNST diagnosis. The Ki67-labeling index is essential for the assessment of NF1 patients. Ki67 levels of 2–5% are seen in ordinary and atypical neurofibromas, while levels of >10% may indicate MPNST [31]. Furthermore, it has previously been reported that CD34-positive cells are diminished in high-grade MPNST [85]. Although the underlying mechanism is unknown, the CD34-positive stromal component may play a role in the formation of MPNST [86]. Nestin is an intermediate filament protein that is stained strongly in the cytoplasm of MPNST [87]. It is more sensitive than other neural markers in the diagnosis of MPNST. Still, it is not easy to distinguish MPNST from desmoplastic melanoma by this marker because they both highly express nestin [87]. Nestin expression has also been found in Schwannoma and neurofibroma [87]. Melan-A, MITF, andHMB45 can aid in distinguishing MPNST from carcinomas and melanomas. GFAP, CD57 (Leu7), and collagen IV are Schwann cell markers, but they have low sensitivities and specificities for MPNST [31].

In recent years, several new MPNST markers have been used as a step forward in the study of MPNST mechanisms. As mentioned earlier, CDKN2A mutation is considered an early-stage mutation for MPNST. Thus, complete loss of the CDKN2A-encoded cell cycle regulator p16 is a common finding in MPNST [31]. However, the loss of p16 does not prove malignancy [31]. The p53 protein often accumulates in tumor cells due to its deregulation or mutation in malignant lesions. As previously stated, the TP53 mutation may play an important role in NF1-associated or sporadic MPNST, and it appears to be a marker of high tumor grade [88]. However, its positivity can also be found in cellular schwannomas or other malignant tumors, leading to misdiagnosis in some cases [89]. p27 is a multifunctional cyclin-dependent kinase that inhibits cell proliferation while promoting cell apoptosis [88]. Zhou et al. reported that nucleocytoplasmic p27 staining was not seen in PN or low-grade MPNST, but in 33% of high-grade MPNST [90]. However, p53, p16, and p27 have not been systematically tested as reliable MPNST markers.

The most recently discovered MPNST marker is H3K27me3. PRC2 has recently been identified as the decisive mutation in the transition from ANNUBP to MPNST. The complete loss of H3K27me3 in immunohistochemical staining is observed in MPNST, with a frequency of 30–90%. It is more common in sporadic and radiation-associated MPNSTs than in NF1-associated MPNSTs [25,91]. The loss of H3K27me3 may also be associated with a decreased chance of survival in MPNST [91]. However, the loss of H3K27me3 cannot distinguish MPNST morphological mimicker synovial sarcoma or fibrosarcomatous dermatofibrosarcoma protuberans, both of which have H3K27me3 losses [91]. A recent study demonstrated that using the technology of methylome-based unsupervised hierarchical clustering, information on DNA copy number profiles, and methylation profiles can be analyzed to differentiate MPNST and BPNST while also assisting in the classification of MPNST [92].

**Table 1 cancers-15-01077-t001:** Immunohistochemical markers for MPNST. This table illustrated the commonly used markers and their expression rate in MPNST.

Markers	Positive Rate	Reference
S-100	50–60%	[82]
SOX-10	27%	[84]
Ki67	>10%	[31]
Nestin	91% (3+)	[87]
p16	45%	[93]
p53	21%	[88]
p27	33%	[90]
H3K27me3	30–90% complete loss	[91]

## 5. Treatment

### 5.1. Surgery

There are limited treatment options for MPNST, and the only effective treatment is complete surgical resection to achieve negative margins [94]. According to data, gross-total resection results in lower recurrence and higher five-years survival rates. Additionally, patients with negative tumor margins have a relatively higher survival rate than those with positive margins [94]. Although the recurrence rate remains very high and post-operative morbidity is significant, the advantages of an aggressive surgical approach outweigh the disadvantages. It is worth noting that NF1-associated MPNST typically develops from preexisting neurofibromas. ANNUBP has better surgical resectability and therapy response than MPNST. However, there is a lack of predictors to determine the timing of surgery due to poor knowledge of the mechanisms of malignant transformation in MPNST.

### 5.2. Chemotherapy

Chemotherapy is an alternative option for those with unresectable or metastatic MPNST. According to the SARC006 prospective study, chemotherapy with adriamycin and ifosfamide resulted in a minimal response, whereas sporadic MPNST responded better than NF1-associated MPNST [95]. Doxorubicin is a first-line treatment for MPNST, and Kroep et al. reported that a doxorubicin–ifosfamide combination regimen provided the best response in MPNST [96]. Several studies have shown that using adjuvant chemotherapy in MPNST treatment has no effect on survival or recurrence rate [97], and its toxicity must be considered. Despite the widespread use of adjuvant chemotherapy for MPNST treatment, its efficacy remains debatable [98]. Other studies have confirmed the ineffectiveness of adjuvant chemotherapy in treating MPNST, except for epirubicin and ifosfamide, which increase the median survival time from 45 to 75 months after local therapies (amputation, wide resection followed by radiation or pre-operative radiation followed by surgery) [99].

### 5.3. Radiation Therapy

Radiation therapy is often recommended for high-grade lesions or tumors larger than 5 cm [100]. The long-term outcome of radiation therapy results in excellent local control [101,102]. However, adjuvant radiation therapy is not beneficial in MPNST survival, even though, in some studies, adjuvant radiation has been used to reduce the tumor size in order to make surgery possible [103]. Brachytherapy and intraoperative electron radiation therapy have also been used in MPNST therapy. According to the Wong et al. study, the five-year local control was 88% in patients treated with brachytherapy and 51% in those treated with external beams [104]. A cumulative dose of ≥60 Gy was required to provide local disease control [104]. Brachytherapy combined with external beam radiation may be more effective [105].

### 5.4. Targeted Therapies

Targeted therapy is the way forward for patients with unresectable or metastatic MPNST. Several clinical trials for targeted therapies have been conducted as our understanding of the molecular pathogenesis of MPNST has increased (Table 2). EGFR inhibitor is one of the earliest attempts at MPNST therapy, as preclinical studies demonstrated that NF1/p53 murine MPNSTs in vitro are stimulated by EGF and inhibited by EGFR inhibitors [106]. However, in a 22-month phase II clinical trial of 24 patients, 19 of 20 evaluable patients did not respond to erlotinib (an EGFR inhibitor), and six patients experienced grade 3 toxicities [107]. Subsequently, in the phase II clinical trial, sorafenib (an inhibitor of Raf kinase and receptor tyrosine kinase) also showed little response to MPNST [108]. Imatinib [109] and dasatinib [110] were ineffective in the treatment of MPNST according the subsequent studies.

As mentioned above, mTOR is an important signaling pathway in MPNST development. In vitro studies shows that mTOR inhibition by everolimus has anti-tumor activity in MPNST cell lines [111]. Johansson G et al. found that everolimus can transiently delay tumor growth in subcutaneous cell-line derived xenografts [52]. In the SARC016 clinical trial, the combination of bevacizumab and everolimus achieved a clinical benefit rate (CBR, the number of patients experiencing a complete response (CR), a partial response (PR), or stable disease (SD) for ≥4 months) of 12% which is considered ineffective in the trial [53,112]. A phase II clinical trial SARC023 (a combination of ganetespib and everolimus) showed no response [113]. In contrast, a preclinical study shows that the Hsp90 inhibitor (IPI-504) combined with the mTOR inhibitor rapamycin can dramatically shrink MPNST in a mouse model (*Nf1/p53* model) [114].

**Table 2 cancers-15-01077-t002:** Registered clinical trials for targeted therapy of MPNST.

Drug	Target	Phase	n	Result	Reference
Erlotinib	EGFR	II	24	no responses, one stable disease	[107]
Sorafenib	RAF VEGFR/c-KIT	II	12	no responses	[108]
Imatinib	c-KIT PDGFR VEGFR	II	7	no responses, one stable disease	[109]
Dasatinib	c-KIT c-SRC	II	14	no responses	[110]
Bevacizumab/Everolimus	VEGF mTOR	II	25	CBR 12% (two stable disease, one partial response)	[112]
Ganetespib/Everolimus	Hsp90 mTOR	I/II	20	no responses	[113]
Selumetinib/Sirolimus	MEK/mTOR	II	21	enrolling	N/A

Summary of previous and ongoing clinical trials. EGFR, epidermal growth factor receptor; RAF, rapidly accelerated fibrosarcoma; VEGFR, vascular endothelial growth factor receptor; c-KIT, stem cell factor receptor; PDGFR, platelet derived growth factor receptor; c-SRC, cellular SRC kinase; VEGF, vascular endothelial growth factor (ligand); mTOR, mammalian target of rapamycin; Hsp90, heat shock protein 90; CBR, clinical benefit rate, (number of patients experiencing a complete response (CR), a partial response (PR), or stable disease (SD) for ≥4 months).

MEK inhibitors are effective in preclinical studies. Trametinib treatment has been shown to reduce tumor growth in MPNST murine models (*Nf1*^flox/ko^;*lox-stop-loxMET*^tg/+^;*Plp*-*creERT*^tg/+^) [115]. Mirdametinib demonstrated a strong reduction in tumor growth with prolonged survival in mouse models [116]. However, there are only a few case reports of MEK inhibitors used in clinical MPNST treatment. A 14-year-old female MPNST patient received trametinib and experienced a sustained response lasting more than 15 months [50]. However, in the therapy of PN, MEK inhibitor selumetinib shows a 72% of response rate in 50 children with PN [117]. Trametinib can induce PN shrinkage and allowing for surgery [118]. Given that PN is a precancerous lesion of MPNST, preventive therapy using MEK inhibitors in PN patients is a good strategy. A promising SARC031 clinical trial using MEK inhibitor selumetinib in combination with the mTOR inhibitor sirolimus is being conducted, but no results have been published yet.

*BRAF* V600 is a novel target for MPNST therapy. Vemurafenib is a selective kinase inhibitor for *BRAF* V600 [119]. Kaplan first reported on a 51-year-old female MPNST patient with the BRAF V600 mutation who received vemurafenib for four days. Following the therapeutic regimen, the tumor size had shrunk by approximately 50% [120]. Although there are no clinical trials of Vemurafenib, it may be effective given the high prevalence of BRAF V600 mutations.

## 6. Prognosis

MPNST has a poor prognosis on average. Previous studies have shown that the five-year overall survival rate is 50—60% [5,24,121,122,123], and the median survival of MPNST is six years [123]. The identification of prognostic predictors is necessary for accurate diagnosis and treatment selection. The French Federation of Cancer Centers Sarcoma Group (FNCLCC) grading system is recommended for MPNST. It is a three-tiered system that evaluates tumor cell differentiation, mitotic activity, and extent of necrosis [81]. Most MPNSTs present as high-grade sarcomas [81], with only 10–15% presenting as low grade sarcomas [124], with the latter thought to be due to the intermingling of MPNST with pre-pseudoneurofibroma [31]. Recent studies have revealed a strong correlation between high FNCLCC grade MPNST and survival outcomes [125,126]. In contrast, the low-grade MPNST was rarely studied. The subjectivity of the grading system could be a challenge for low and intermediate grade lesions [125]. American Joint Committee on Cancer (AJCC) STS staging system is another mixed clinical–pathologic system. It is a four-tiered system based on STS tumor size, depth, grade, and the presence or absence of metastasis [83]. However, it also has disadvantages in that some non-metastatic lesions may meet the criteria of level III/IV under the MPNST setting [83]. A consensus on staging system is required to better assess the prognosis of MPNST.

In addition to grading systems, some independent predictive factors of prognosis have been reported in various series. NF1 mutation is associated with worse survival than sporadic MPNST [24,123,127]. Further studies are needed as the optimal treatment regimen for NF1-associated, and sporadic MPNST may differ. Age over 60 years was also considered an independent predictor, although it was rarely reported [123]. Some studies show that tumors larger than 5 cm significantly reduced disease-specific survival (DSS) [5] and overall survival (OS) [123], whereas the tumor depth is only reported in one study [121]. The influence of different therapeutic methods is under investigation. As previously stated, resectability is important in MPNST patient therapy. R1 and R0 resections had significantly better survival rates than R2 resections. The value of radiotherapy and chemotherapy for MPNST survival is still being debated.

## 7. Challenges and Prospects

Many questions remain to be investigated in the future. Firstly, as the mechanism of sporadic MPNST is not fully characterized, the differences and relationships between NF1-related MPNST and sporadic MPNST are not well explained. Further exploration is needed to assess the risk of both types of MPNST in a better way and select appropriate treatment strategies. Secondly, the specific diagnostic criteria for MPNST must be defined. Although MPNSTs can be roughly assessed by imaging and immunohistochemistry in clinical practice, more precise criteria are still needed for more detailed classification and grading of MPNSTs. Detection techniques using genetic and molecular probes could be a future direction.

Complete surgical resection remains the most effective treatment for MPNST. A combination regimen of chemotherapy, radiation, and targeted therapy may achieve better survival, but more consensus is needed and toxicity should also be carefully assessed. Additionally, research on MPNST-targeted drugs is an important development goal. Currently most of drugs in clinical trials are based on the RAS and tyrosine kinase receptor pathways. However, the majority of them have failed clinical trials. Recent MPNST research has revealed that complex PRC2 mutations and H3K27me3 loss play a critical role in MPNST development. Using gene therapy to target CDKN2A or NF1 could be a novel breakthrough [79]. However, the premise of the above research is to develop a better in vitro and in vivo model. The current animal models, for example, the most widely used cisNf1+/−p53+/− model, are difficult to fully simulate the entire process of MPNST development because MPNST is a complex disease with multiple genes involved. Establishing pathological changes in animal models similar to human changes is difficult, especially in precancerous PN and ANNUBP [128]. Moreover, gene expression in animal models differs significantly from that in humans [128]. The main issue for clinical research, is the rarity of MPNST The clinical and genetic data for MPNST are insufficient due to a lack of comprehensive and extensive application of gene sequencing technology. However, these are expected to be remediated in future research. The prognostic factors of MPNST may require more detailed analysis for precision medicine, such as comparing the prognosis of people with different genetic mutations.

## 8. Conclusions

Significant breakthroughs have recently been made in the study of MPNST. On the one hand, risk factors of NF1 and radiation exposure are further explained in MPNST development. The precancerous lesions of PN and ANNUBP have revealed the genetic and molecular mechanism of MPNST. On the other hand, there has been a greater focus on diagnosing and treating MPNST, particularly novel molecular biomarkers and combined therapy regimens.

## Figures and Tables

**Figure 1 cancers-15-01077-f001:**
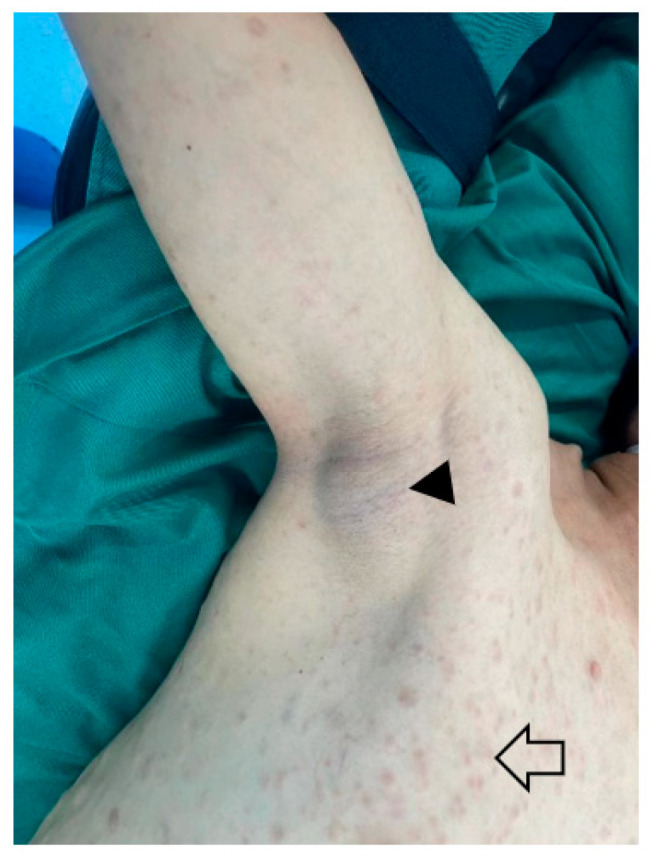
A 41-year-old male with Cafe-au-lait macules of NF1-associated with MPNST. The macules spread throughout the body (**⇦**), and MPNST is under the skin of the right axilla (▲).

**Figure 2 cancers-15-01077-f002:**
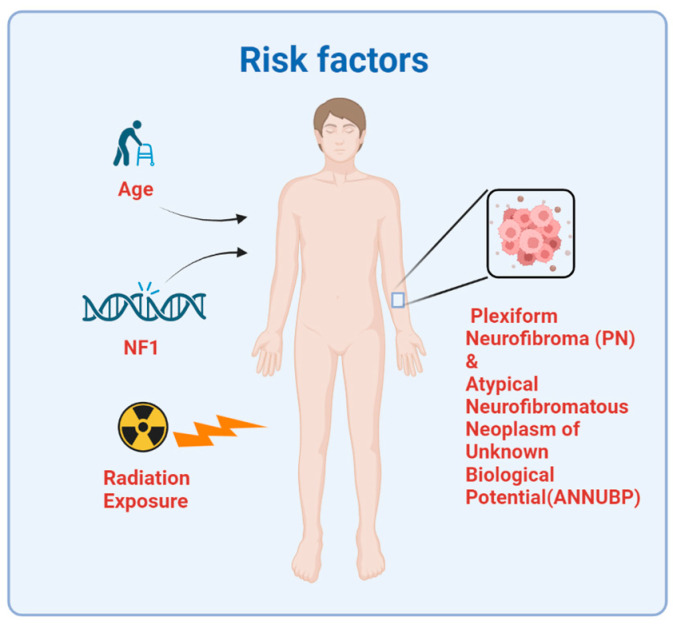
Risk factors of MPNST. MPNST often occurs in patients aged 30–50. About 50% of MPNST cases are associated with NF1, and about 10% of MPNST patients have a history of radiation exposure. The possession of PN and ANNUBP has malignant potential for MPNST development.

**Figure 3 cancers-15-01077-f003:**
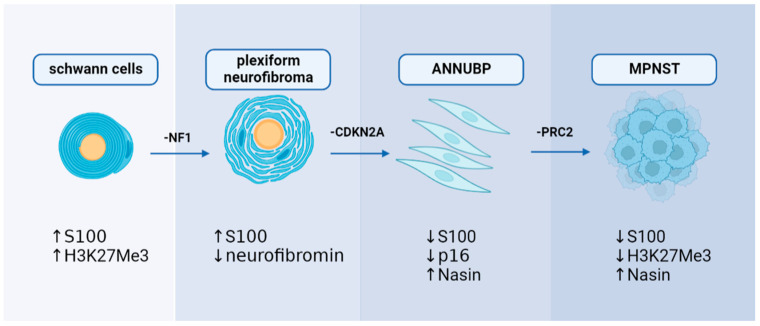
A hypothesis of sequential, multiple-hit genetic mutations in MPNST development: *NF1*, *CDKN2A*, and *PRC2* may play a decisive role in MPNST development with alterations in immunohistochemical markers.

**Figure 4 cancers-15-01077-f004:**
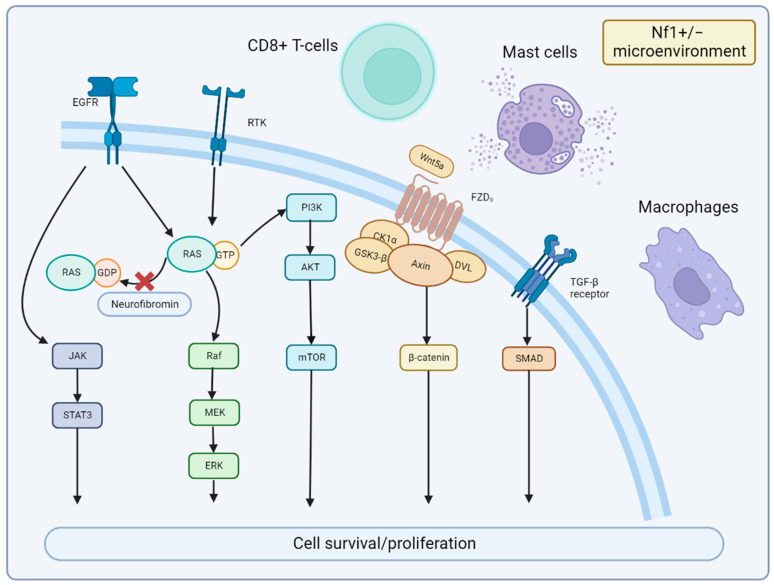
Signaling Pathways and Microenvironment of MPNST. Neurofibromin encoded by NF1 is lost in MPNST, which leads to the continued activation of RAS signaling and downstream MEK and mTOR effectors. JAK/STAT3 and Wnt/β-catenin signaling are also involved in MPNST development. Nf1+/− microenvironment of mast cells, CD8+ T cells, and macrophages can promote MPNST growth with cytokines or immune dysfunction.

**Figure 5 cancers-15-01077-f005:**
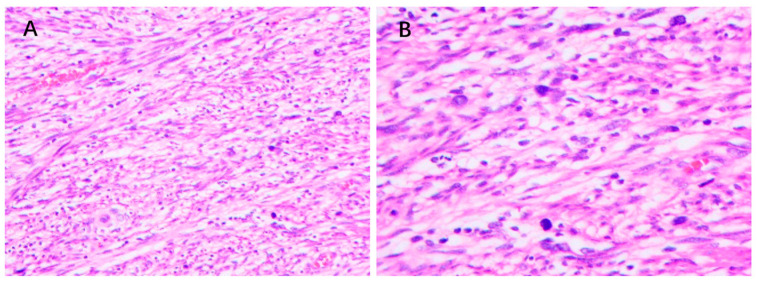
Histological appearance of MPNST. hematoxylin-eosin staining for MPNST (**A**) magnification 200× (**B**) magnification 400× The cells are elongated into a spindle-like shape with a hyperchromatic nucleus. Cell necrosis and mitosis are common in MPNST.

**Figure 6 cancers-15-01077-f006:**
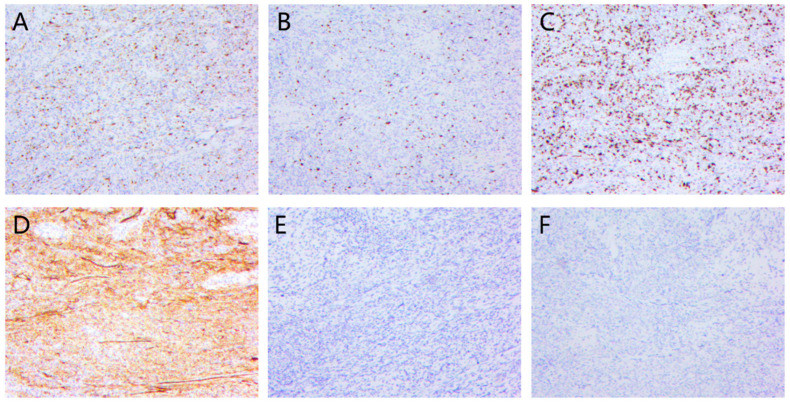
Immunohistochemical appearance MPNST. (**A**) S100 protein-positive cells are largely reduced in MPNST. (**B**) SOX-10 protein-positive cells are fewer than S100. (**C**) Ki-67 labeling index is elevated to >10%. (**D**) CD34-positive fibroblastic lattice is reduced. (**E**) Desmin stain is negative. (**F**) H3K27me3 is completely loss in MPNST.

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
