# Peer review of "Malignant Peripheral Nerve Sheath Tumors: Latest Concepts in Disease Pathogenesis and Clinical Management"

_cancers, 2023, doi:10.3390/cancers15041077_

Round 1

Reviewer 1 Report

General Comments:

This is a comprehensive review of MPNST pathogenesis, diagnosis, treatment, and future challenges and prospects. The review is very detailed and will eventually be valuable to the journal readership.

Requested revisions:

The figures are located in appropriate positions in the manuscript, but none of the figures are referenced in the text of the manuscript. Figure 1 is referenced in appropriately on line 146.

The article requires extensive English language editing for grammar. I would suggest final review by an experienced English language scientific editor prior to resubmission to the journal. I have tried to indicate areas where the grammar is most problematic or requires the most attention.

The names of genes need to be italicized throughout the entire manuscript when referring to the gene (e.g. NF1 deletion, TP53 mutation, gene loss of SUZ12) and not the protein.

I will note concerns according to their position in the manuscript:

Line 3: I recommend changing the title of the manuscript to “Malignant peripheral nerve sheath tumors: latest concepts in disease pathogenesis and clinical management”

Line 40: define the term “WHO” (World Health Organization). The sentence should read: “MPNST was first classified as a soft-tissue sarcoma by the World Health Organization (WHO) in 2013…”

Line 46: The manuscript states that the age of onset is around 30-50 years, while NF1 patients may occur 10 years earlier. In fact, it can occur even earlier than this. I would rephrase this sentence. The sentence should read: “The age of onset is around 30-50 years, while in NF1 patients it may occur 10 years earlier on average.”

Line 67 – define what a Lisch nodule is in the manuscript.

Line 71 – NF 1 has a space here. Be consistent throughout the manuscript with NF1 (no space)

Line 72-74 - The sentence should read: “With the new development of analysis techniques such as multiplex ligation-dependent probe amplification (MLPA), comparative genomic hybridization (CGH) array, and next generation sequencing (NGS), NF1 genotype-phenotype correlations are being elucidated.”

Lines 76-80 – The caption to Figure 1 needs to be rewritten for grammar.  Figure 1 needs to be referenced in the paragraph somewhere for section 2.1

Line 81 – Define the study population for the risk of 0.06% rate of post-radiation sarcomas. Is this in cancer survivors? Cancer survivors of a certain type?

Line 81-82 – This sentence “As for MPNST…” needs to be rewritten for grammar. The sentence should read: “Approximately 10-13% of MPNST patients have a history of therapeutic irradiation.”

Line 88-89 – This sentence needs to be rewritten for grammar: “it may have the common mechanism as sporadic MPNST” doesn’t make sense.

Line 94-95 - The sentence should read: “ANNUBP and PN are considered precancerous lesions and can therefore help illustrate the pathogenesis of MPNSTs.”

Line 99 – What is meant by “them” in this sentence? Is it talking about ANNUBP and PN?

Line 101 – “Some patients remain asymptomatic throughout their lives.” Is this talking about patients with the precancerous lesions ANNUBP and PN?

Line 102-104 – This sentence needs to be rewritten for grammar

Line 109-113/Figure 2 – Needs to be referenced in the manuscript text. I would consider using the full words for PN and ANNUBP in the graphic rather than the abbreviations since it won’t take up any more space here. The Figure 2 legend needs to be rewritten for grammar.

Line 114 – I recommend renaming this section: “Mechanisms of MPNST pathogenesis”

Lines 114-146 and Figure 3 and multiple other areas in the manuscript – the abbreviation PCR2 Polycomb repressive complex 2 is not correct. The correct abbreviation is “PRC2”. PRC2 is a protein complex that arises from proteins coded by multiple genes. Therefore, it is not appropriate to say PRC2 mutation – you will need to say “mutation in genes that code for PRC2 complex proteins” or something similar.

Lines 114-146 – I would use the term “H3K27 global hypomethylation” rather than “complete loss” of H3K27me3 because that is not accurate. There are many grammatical errors in this section. The phrase “increase in CDKN2A/B deficiency” is difficult to understand. “Benign neurofibroma” should not be capitalized. Again, italicize gene names if you are referring to the gene and not the protein.

Line 151 – “Several studies have also mentioned” – this language sounds too casual for a manuscript and I suggest this sentence be rewritten.

Line 153 – Do you mean that TP53 mutation may an independent prognostic factor for MPNST?

Line 152 – Instead of saying “A study illustrates…” perhaps you can use the name of the authors et. al. here as you have done in other places in the manuscript.

Line168 – The term research hotspots is not appropriate – perhaps use “important areas of future research focus” or something like that.

Line 169 “It can promote cell l proliferation” What is “it” referring to here? 

Line 182 – EGFR-STAT3 are also “reported in MPNST” – Reported to do what?

Line 190-195 – Figure 4 needs to be referenced in the text. I would capitalize the term “Mast cells” to match the capitalization of the other cell types shown in the figure.

Line 198-200 – This is a run on sentence that should be rewritten.

Line 202 – “Gene mutation” should not be capitalized

Line 207-209 – This sentence needs a reference.

Line 216 - The sentence should read: “…but further validation is still needed.”

Line 217 – The abbreviation BPNST – is this defined in the manuscript? Is it necessary to use an abbreviation for this since it isn’t used that many times?

Lines 218-226 – The terms ADCmin, DWI, ADC, SUV, SUVmax all need to be defined in the manuscript.

Lines 236-241 – The sentence about biopsy causing pain and have risks for metastasis needs to be eliminated or highly referenced. Since the differential is so broad for these tumors, biopsy is likely the only way you know what you are working with. Also, the risk of metastasis related to biopsy is not established to my understanding.

Line 255 – Eliminate the words “from these variants”

Line 256 – Consider changing “a special role” to “an important role…”

Figure 5 – Needs to be referenced in the text. I suggest a higher power image to the right of the existing image to better demonstrate the hyperchromatic nuclei. I think you can eliminate the biological sex and age of the patient from the figure caption because it is not relevant to the manuscript.

Line 267 – Schwan-associated should be “Schwann-associated”

Line 299-300 – The information about p27 requires a reference.

Line 302 – The phrase “it probably could be a marker in differentiation” should be eliminated.

Table 1 – Make sure Table 1 is referenced in the manuscript text.  Nestin is the correct word, not “Nastin” This needs to be corrected throughout the manuscript. What does expression rate mean in this table? Does expression rate refer to the proportion of tumors that stained positive for the respective biomarkers in each of the studies?

Figure 6/Lines 315-318 – This figure needs to be referenced in the text. I think that I would eliminate the biological sex and age of the patient because they are not relevant to the manuscript as best I can tell.

Line 329: The sentence should read: “However, there are a lack of predictors…”

Line 333 – I would not use the term “regular chemotherapy”. I would eliminate the use of “regular.”

Line 336 – Would replace J. R. Kroep with Kroep et. al if there are multiple authors.

Line 340-343 – This sentence needs to be rewritten for grammar.

Line 355 – I would call this section “Targeted therapies.” In this section I would use “targeted therapy” instead of “target therapy” or “targeting therapy”

Line 365 – The term “proved useless” is a bit harsh. The sentence should read: “Imatinib and dasatinib were ineffective in the treatment of MPNST.”

Line 366 – The sentence should read: “mTOR is an important signaling pathway in MPNST development”.

Lines 369 – are these patient-derived xenografts or derived from cell lines?

Line 370 – a “clinical benefit rate” needs to be better explained or defined. I do not believe the reader will be familiar with this term. This is also used in table 2.

Line 374 – Specify the type of mouse model here and whenever a mouse model is mentioned throughout the manuscript.

Table 2 – Make sure Table 2 is referenced in the manuscript text. Keep the title of the table above the table and move the remainder of the table legend to below the table.

Line 398 – When the FNCLCC is first introduced, a reference should be provided.

Line 404- The classification that is “not useful” is unclear. Is this the FNCLCC classification?

Line 410 – Independent predictive factors of what? Prognosis?

Line 425 – “The mechanism of sporadic MPNST is still unknown” – in fact this manuscript details several possibilities or findings. Perhaps it would be better to say “The mechanism of sporadic MPNST is not fully characterized.”

Line 441 – “The current animal model is difficult” – What animal model? Be specific.

Line 463 – Should say “Funding” and not “Fundings”

Line 468 – This should say “Competing Interests” or “Conflicts of Interest” depending on the journal format  and not Competing of interests”.

Line 469 – This is a review article and there should not be any primary data sets associated with the review. The data and materials section therefore does not make any sense.

Reviewer 2 Report

The Authors should be acknowledged for their comprehensive review on MPNST. In their review, they provide a thorough overview of MPNST molecular and pathological characteristics, with hints concerning main treatment strategies.

A few , minor comments. 

Table 1: "nastin" should be substituted with "nestin"

Figure 3: "nasin" does not exist. please correct. 

5.2. Chemotherapy: "regular" (which is not a definition) chemotherapy for sarcomas entails doxorubicin and ifosfamide, not cetuximab, please correct.

Sometimes the meaning of sentences is impaired due to English language, this should be extensively revised.

Reviewer 3 Report

This is a general review regarding the epidemiology, diagnosis and treatment of MPNST. I have a number of comments that will make this a stronger manuscript:

1. Although the paper covers a wide variety of topics, there are recent important developments that could be added. Some of this include:

1.1. Genetics and targeted therapy: A subset of sporadic MPNST have been shown to harbor BRAF V600 and NRAS Q61 mutations, that may be amenable to targeted treatment (with BRAF-mutant cases being anecdotally responsive to vemurafenib and, as the authors mention, existing clinical trials with MEK inhibitors). See PMID: 3606782924335681, 30099373.

1.2. Methylation changes in MPNST are a recent development and show differences between low grade MPNST, epithelioid MPNST and conventional high grade MPNST, among others. This topic is worth discussing, and ties in with other molecular findings. Please see PMID: 26857854

1.3. Prognosis: Histologic grading of MPNST has been controversial in the past, as the authors mention. However, more recent studies have shown a good correlation between high FNCLCC grade and survival outcomes, and low grade lesions tend to behave in a fairly indolent fashion. PMID: 34962906, 33880447, 31078939.

2. Histologic images: The image of the H&E-stained slide is out of focus and not the best example of the characteristic histologic features of MPNST. I would recommend adding additional photos in that panel and include a Bone and Soft Tissue Pathologist from the authors' institution. 

2. Style changes: Please remember that human genes should be written in upper caps and italicized. 

Round 2

Reviewer 1 Report

Overall, the revisions are satisfactory. There are still minor/moderate English language grammar edits that need to be made. However, after English language review one more time, I recommend this manuscript be published.

Author Response

​Thanks so much for the recommendation. We sent our manuscript to a company for proofreading and English editing. We hope the latest version will be satisfactory.

Reviewer 3 Report

Most/all suggestions have been addressed. 

Author Response

Thank you very much. We sent our manuscript to a company for proofreading and English editing. We hope the latest version will be more satisfactory.